# Auranofin Targeting the NDM-1 Beta-Lactamase: Computational Insights into the Electronic Configuration and Quasi-Tetrahedral Coordination of Gold Ions

**DOI:** 10.3390/pharmaceutics15030985

**Published:** 2023-03-18

**Authors:** Iogann Tolbatov, Alessandro Marrone

**Affiliations:** 1Institute of Chemical Research of Catalonia (ICIQ), The Barcelona Institute of Science and Technology, Av. Paisos Catalans 16, 43007 Tarragona, Spain; 2Dipartimento di Farmacia, Università degli Studi “G. D’Annunzio” Chieti-Pescara, Via dei Vestini 31, 66100 Chieti, Italy

**Keywords:** auranofin, NDM-1 metallo-beta-lactamase, DFT calculations, Zn/Au exchange, metal coordination

## Abstract

Recently, the well-characterized metallodrug auranofin has been demonstrated to restore the penicillin and cephalosporin sensitivity in resistant bacterial strains via the inhibition of the NDM-1 beta-lactamase, which is operated via the Zn/Au substitution in its bimetallic core. The resulting unusual tetrahedral coordination of the two ions was investigated via the density functional theory calculations. By assessing several charge and multiplicity schemes, coupled with on/off constraining the positions of the coordinating residues, it was demonstrated that the experimental X-ray structure of the gold-bound NDM-1 is consistent with either Au(I)-Au(I) or Au(II)-Au(II) bimetallic moieties. The presented results suggest that the most probable mechanism for the auranofin-based Zn/Au exchange in NDM-1 includes the early formation of the Au(I)-Au(I) system, superseded by oxidation yielding the Au(II)-Au(II) species bearing the highest resemblance to the X-ray structure.

## 1. Introduction

Bacterial resistance regularly hampers the efficacy of therapy with subsequent grave repercussions, most notably in old or seriously sick persons. It is driven by inadequate empiric antibacterial treatment, characterized as the early employment of an antibacterial medicine to which the microorganism is not vulnerable and spuriously protracted treatment with antimicrobials [1].

One of the most immediate menaces is carbapenem-resistant Enterobacteriaceae, the presence of which in human bloodstream results in death in almost half of cases [2]. These microorganisms bear metallo-β-lactamases (MBLs), for example, New Delhi metallo-β-lactamases (NDMs), which give the resistance against the β-lactams including cephalosporins, penicillins, and carbapenems, i.e., the most frequently used class of antibiotics, particularly for the treatment of serious Gram-negative bacterial infections [3]. Resistance to β-lactams is provided by plasmids bearing MBLs; moreover, the ease with which these plasmids are transferred between various species results in their universal spreading [4]. That is why the synthesis of broad-spectrum MBL inhibitors is of utmost importance.

MBLs are zinc metalloenzymes, which feature one or two zinc ions Zn(II) and the nucleophilic hydroxyl (OH^−^) in between, which plays a crucial role in the hydrolysis of the β-lactam ring, thus disrupting the antibiotic’s action [5]. MBLs are characterized by a great structural diversity; hence, the formulation of a broad-spectrum inhibitor for all of them is problematic. The key part played by Zn(II) ions makes them a perfect target for the tentative inhibitors. Indeed, it was shown in numerous studies that Zn(II) ions are crucial for the resistance of MBLs to antibiotics since these metal cofactors are involved both in the substrate binding and catalysis [6,7,8]. NDM-1 is a member of class B1 of MBLs, the typical feature of which is the presence of two metal centers [3]. Despite ongoing dispute on the roles of Zn metal centers in B1 MBLs, there is a general agreement that the role of Zn1 is in anchoring OH, whereas Zn2 is coordinated by C3 (penicillins)/C4 (cephalosporins) carboxylate present in the 5/6-membered ring fused with the β-lactam ring of these antibiotics, thus providing stabilization for the anionic intermediate (Figure 1) [9,10,11].

Obviously, the chelation of zinc by ligands such as aspergillomarasmine A [13], or its replacement by another metal, for instance, by Bi(III) from bismuth citrate [14] or Au(I) from auranofin [15], incapacitates the complex NDM-1 machinery directed at the β-lactam breaking.

Auranofin (AF, 1-thio-β-D-glucopyranosatotriethylphosphine gold-2,3,4,6-tetraacetate) was the first orally administered gold-based metallodrug developed specifically to remedy rheumatoid arthritis, certified by the US Food and Drug Administration (FDA) in 1985 (Figure 2) [16]. Its therapeutic effect is ascribed to the [AuPEt_3_]^+^ cation yielded after the detachment of the thiosugar in the biological milieu [17]. This cation has an augmented preference to thiol- and selenol-based proteins [18], thus causing the disruption of cellular metabolism pathways and leading to the intended medicinal impacts.

In a recent study, auranofin was recognized as an inhibitor of MBL NDM-1 that completely and irreversibly blocks the enzyme function, substituting zinc ions in the enzyme’s active site [15]. The authors demonstrated that the administration of auranofin resensitizes carbapenem- and colistin-resistant bacteria to antibiotics and slows down the development of β-lactam and colistin resistance. It was found that both Zn(II) ions are substituted by the Au(I) ions delivered by auranofin and that two gold ions replace both Zn1 and Zn2 in the active site of NDM-1 by assuming a quasi-tetrahedral coordination geometry [15]. The X-ray structure of the NDM-1 active site with Zn1/Zn2 replaced by two gold ions is shown at Figure 3. The coordination geometries of gold ions are rather unusual.

Although the preferred geometry of gold(I) complexes is linear with two-coordination at the gold center, instances of tetrahedral coordination have been reported [19]. Solution NMR studies have demonstrated that the addition of excess phosphine to [AuL_2_]^+^ (L = phosphine) can result in the formation of the [AuL_3_]^+^ and [AuL_4_]^+^ species, but unexpectedly, few of these complexes have been structurally characterized [20]. Furthermore, the majority of the four-coordinate species that have been identified comprise either bidentate phosphines [21] or thiolate ligands [22].

Taking into account that the gold is presented in the form of Au(I) in auranofin, with a distinct linear bicoordination, the detected tetrahedral coordination of both gold ions deserves to be further clarified. One explanation might be the occurrence of either oxidative or reductive processes accompanying the Zn(II)/Au(I) exchange. On the other hand, the NDM-1 fold itself might template the quasi-tetrahedral coordination and constrain the two Au(I) ions to adopt the tiled coordination geometry.

The concomitance of both protein constraint and redox processes would be another reasonable explanation for the observed gold coordination in the NDM-1 active site. In this frame, we thus emphasize two aspects of extreme importance that need to be elucidated. On one hand, an in-depth understanding of the mechanism of the Zn/Au exchange, particularly the chemical steps underlying the decomplexation of the strongly coordinated PEt_3_ ligand from the [AuPEt_3_]^+^ cation, would shed more light onto the biological activity of auranofin. Indeed, although there is a general consensus that the activation of auranofin yields the cationic complex [AuPEt_3_]^+^, the mechanism of its action in the active sites of metalloenzymes is not yet fully understood [23,24]. The majority of authors agree that the sulfhydryl and selenohydryl selectivity for gold controls the process of auranofin’s gold dismantling its ligands via the substitution reactions with either Cys and Sec protein residues or with the free thiols in the cytoplasm [25]. That is why there exists a long-standing debate on whether auranofin loses firstly its ligands in the cytoplasm via substitution by various thiols and only after a series of ligand exchanges reaches its intended biomolecular target, or whether auranofin’s activation happens only after it reaches its target. Therefore, the experimental investigation of the chemical processes occurring after the administration of auranofin is very challenging due to the high chemical complexity impacted by this metallodrug either before or after its entry in the targeted cell. Nevertheless, the theoretical chemists have not yet succeeded in responding to this question either, despite numerous computational articles focused on the disentanglement of the mechanistic process of auranofin’s ligand exchange reactions [17,26,27,28]. Moreover, to the best of our knowledge, all the available computational studies of auranofin have addressed the behavior of one auranofin complex with its intended targets, for the sake of simplicity. The assessment of plausible mechanisms of the Zn/Au exchange operated by the attack of the [AuPEt_3_]^+^ cation to the active site of protein NDM-1 is currently the focus of our ongoing theoretical investigation. Indeed, computational studies were often successfully utilized for the analysis of the interaction of metal ions and metallodrugs with proteins [29,30].

Another aspect of high importance emerging from the X-ray detection of the Zn/Au exchange in the NDM-1 active site is represented by the unusual tetrahedral coordination of the two gold ions. Indeed, despite the fact that the oxidation of Au(I) to Au(III) is accompanied by the increase of the coordination number from two to four, the resulting coordination geometry is expected to turn from linear to square planar, so the quasi-tetrahedral coordination of the gold centers in NDM-1 appears to be elusive.

In this paper, we address the coordination of Au ions observed in the NDM-1 active site by the use of density functional theory approaches. We extracted a reduced system modelling the coordination of the gold bimetallic scaffold, incorporating two gold centers, their ligands, and the bridging hydroxyl from the available X-ray structure [15], and assumed all possible combinations of charge and multiplicity of this system. The optimization of this system in various charge–multiplicity states and the comparison of the obtained geometries with the experimental results [12,15] allowed us to infer the oxidative states of gold cations in NDM-1 and eventually expand our understanding of the chemical structure of NDM-1 upon the reaction with auranofin. Moreover, we performed the optimization of the gold-bound NDM-1 active core by either freezing or not freezing the Cartesian coordinates of the atoms resembling the alpha carbon atoms of the real enzyme. This strategy allowed us to better assess the role of the NDM-1 backbone in the shaping of the bimetallic scaffold coordination. In addition to providing an interpretation to the detected X-ray data, this theoretical study delivered preliminary insights on the viable mechanisms of the Zn/Au exchange.

## 2. Computational Details

All calculations were performed with the Gaussian 09 A.02 quantum chemistry package [31]. Geometrical optimizations were carried out in solution by using ωB97X [32] in combination with the def2SVP basis set [33,34]. The input geometries of NDM-1 with Zn or Au ions were obtained from the pdb entries 5zgz [12] and 6lhe [15], respectively, by modelling the metal-bound residues with the corresponding side chains, capped with a methyl group resembling the alpha-carbon atoms. To take into account the anchoring of these groups to the NDM-1 backbone, the C atoms of terminal methyl groups were kept frozen during the geometry optimization. Frequency calculations were performed to verify the correct nature of the stationary points and to estimate the zero-point energy (ZPE) and thermal corrections to thermodynamic properties. Despite the presence of artificial restraints during optimization, the computed frequency spectra did not produce any imaginary frequencies. DFT gives a good description of geometries and reaction profiles for complexes formed by transition metals [35,36] and gold in particular [37]. Therefore, the ωB97X functional is known to reach a high accuracy in the calculation of electronic energies [38,39]. The NBO electron spin densities were calculated as the difference of alpha and beta natural electron configurations [40]. The PCM continuum solvent method was used to describe the solvation [41]. The water was used as an implicit solvent due to the location of the NDM-1 active site on the surface of the protein. The solvent-accessible surface (SAS) of NDM-1 was assessed by means of the sas tool in Gromacs [42].

## 3. Results and Discussion

The active site of NDM-1 is characterized by the presence of two zinc metal centers, Zn1 and Zn2, at a distance of 3.62 Å and is connected through a hydroxyl bridge (Figure 3a). The other ligands at Zn1 are three histidines, 120, 122, and 189, that complete the tetrahedral coordination of this center. On the other hand, Zn2 reaches a trigonal bipyramidal geometry through the coordination of Asp124, Cys208, His250, and a water molecule at the distances in the 2.2–2.5 Å range. Such a coordinative asset corroborates previous studies [9,10,11] that have indicated Zn1 as the metal center that affixes the OH in the correct position, whereas the labile water on Zn2 can be replaced by carboxylate substrates, for instance, the C3/C4 carboxylate of the β-lactam antibiotic [43].

The substitution of Zn(II) by Au ions [15] determines appreciable modifications of the NDM-1 active site (Figure 3b,c). X-ray studies have led to two coordinative variants of the bimetallic gold moieties. In one case, two oxygens are bound to the Au1–Au2 scaffold; one oxygen is bound at a short distance (1.95 Å) with Au1, thus presumably being a hydroxyl group, whereas another oxygen is coordinated at Au2 at a longer distance (2.43 Å), thus more likely corresponding to a water ligand. The distance between the two Au centers is 3.76 Å (Figure 3b), which is longer than those detected in various crystallographic studies in the range 2.47–3.49 Å (Appendix A) [44,45,46,47,48,49,50,51]. These data suggest that gold metal ions are not bound and held in place by the surrounding protein residues. Therefore, the formation of metallophilic Au(I)…Au(I) interactions—also denoted as closed-shell d10-d10 interactions—has been indicated to occur in the range 2.75–3.25 Å [52], so the two gold centers observed in the NDM-1 enzyme are not directly bonded.

Besides the two oxygens bound at the Au1 and Au2 metal centers, Au1 is also bonded to the three histidines, 122, 189, and 120, at distances of 2.00, 2.05, and 2.50 Å, respectively, while Au2 is bound to Asp124, Cys208, and His250 at distances of 2.23, 2.47, and 2.27 Å, respectively. Hence, both metal centers present a tetrahedral coordination, which is rather unusual for this transition metal species [53]. We can conclude that in the substitution of the Zn(II) ions with Au, Au1 resembles Zn1 in the coordination of the hydroxyl ligand, even though the latter group does not form a bridge, as detected in the native NDM-1 structure. Therefore, the coordinative bond between Au1 and His120 resulted to be sensibly elongated compared to His122 and His189, thus distorting the tetrahedral-like coordination of Au1.

In the X-ray variant with just one oxygen (Figure 3c), the Au2-O distance of 2.38 Å is consistent with the coordination of a water molecule, while no hydroxyl group resulted coordinated to the bimetallic system. Such a reorganization is rather important because of the role played by the metal-coordinated hydroxyl in the NDM-1-catalysis mechanism. On the other hand, we noticed that the coordination of Au1 shown in Figure 3c also features an increased elongation (by 0.27 Å) of the His120–Au1 bond compared to the system with the bound hydroxyl (Figure 3b). The loose coordination to one of the histidines, i.e., His120, and the lack of coordinated OH^−^ in one X-ray structure (Figure 3c) suggest that the gold metal centers eventually bound to the NDM-1 enzyme may assume different oxidation states; in particular, we envision that at least Au1 may be found in two oxidation states, the higher one able to coordinate the hydroxyl ligand (Figure 3b) and the lower one lacking this anionic ligand and almost unbound to His120.

Based on these evidences, we carried out density functional theory calculations with the aim of better characterizing the bonding structure of the Au1–Au2 bimetallic moiety and providing a preliminary insight of the mechanism of the Zn(II)/Au(I) exchange. For this purpose, we assigned various combinations of charge and multiplicity to the system, incorporating two gold cofactors and all the ligands found within 3.0 Å from the metal centers, i.e., OH, water, Asp, Cys, and four histidines. In particular, we assumed oxidation states of the gold centers 0, I, II, and III and multiplicities assigned consistently with the configurations reported in Table 1. With respect to the oxidation state I of gold in the [AuPEt_3_]^+^ cation, the selected values reflect the occurrence of either oxidation, reduction, or even none of these processes, accompanying the binding at NDM-1. The exploration of different values of multiplicities in some instances (Table 1 and Table 2, Figure 4) was instead performed in consideration of the rather unusual tetrahedral coordination of both the Au1 and Au2 centers, as well as the presence of elongated distances, i.e., His120-Au1, that could be originated by the presence of unpaired electrons on the metal center. Afterwards, we conducted optimizations by starting from the same X-ray input geometry, obtained by either chain A or chain B of the pdb entry 6lhe, in which the positions of the carbon atoms resembling the Cα of the NDM-1 residues were kept frozen. This option allowed us to model the rigid arrangement of the metal-coordinated ligands imparted by the NDM-1 backbone. The assignment of the charge -3 described the system Au(0)-Au(0) (**A**, Table 1). For this bimetallic system with the multiplicity of 1, the calculated intermetallic bond is 4.11 Å, and the coordination of Au1 with two of the three histidines disappears, while all the Au2–ligand distances elongate, thus making such a combination of charge/multiplicity rather unlikely. Moreover, Au(0)-Au(0) system **B** with the multiplicity of 3 was found to lose the coordination picture of the active site detected in the X-ray structure and in addition showed the formation of a strong Au–Au bond of 2.84 Å which is not experimentally detected.

The geometry optimization of Au(I)-Au(I) systems **C**-**E** was carried out by exploring three different multiplicity configurations: 1, 3, and 5, with the corresponding configurations of the two gold centers bearing 0, 1, and 2 unpaired electrons (Table 1). As shown, our calculations show that systems **C** and **D** do not reproduce the bimetallic gold system detected in the crystal structure well, with both Au1 and Au2 reducing their coordination numbers. On the other hand, system **E** with a multiplicity of 5 resembles the experimental structure much better, as shown by the RMSD value of only 0.83 Å (Table 2).

The Au–Au system **F** with a charge of 0 and a multiplicity of 2 leaves two possible oxidative state combinations, i.e., Au(I)-Au(II) and Au(0)-Au(III) (Table 1). In fact, DFT calculations showed that this configuration of the bimetallic system disrupts the experimentally detected coordination of both gold centers; the Au–Au distance is elongated greatly, 7.02 Å, while other coordinative bonds are lost (Table 2, Figure 4).

The DFT optimization of systems **G** and **H** with a charge of +1 and a multiplicity of 3 and, thus, the possible combinations Au(I)–Au(III) or Au(II)–Au(II), respectively, led to the geometry that resembles mostly the X-ray structure of the NDM-1 active site (chain B of PDB 6lhe), with an RMSD value of 0.74 Å (Table 2). In this case, the coordination pattern of the two gold ions resembles the experimental data well, with the only exception being the lack of the coordination between Au2 and Asp124 in system **G** (Table 2). The Au–Au distance for this system was calculated to be about 5 Å; although this value is quite higher compared to the X-ray structure, it correctly corroborates the absence of any Au–Au bond or interaction. Nevertheless, the fact that the coordination of the Au–Au scaffold is correctly assessed by assigning a charge of +1 and multiplicity of 3 in system **G**, but not with the multiplicity of 1 in system **H**—featured an RMSD value of 2.12 Å (Table 2)—is exceptionally well in agreement with the experimental results and suggests that the gold ions, delivered from two auranofin complexes to the metallo-β-lactamase NDM-1, are more likely to be in the Au(II)–Au(II) oxidative state.

The geometry optimization of the bimetallic systems with charges of -1 and +1 was then repeated at the same level of theory by removing any constraints. These calculations helped to elucidate the role played by the protein backbone, on which the metal-coordinating side chains are installed, in shaping the detected coordination of Au1 and Au2; the unconstrained optimization performed on systems **C**–**E** and **H** was found to mostly resemble the experimental architecture (Table 2). As shown, in the absence of any geometrical constraint, the Au(I)–Au(I) systems **C** and **D**, i.e., singlet or triplet, respectively, are both optimized to structures with coordinative bond patterns better resembling the X-ray structure, with the corresponding RMSD values being only 0.91 and 0.80 Å, respectively. On the contrary, the coordination pattern of the more elusive quintet configuration in system **E** was found to deviate majorly in the absence of geometrical constraints. These outcomes seem to corroborate the formation of the Au(I)–Au(I) bimetallic system in the NDM-1 enzyme with either singlet or triplet configurations and indirectly indicate that the replacement of Zn(II) ions by [AuPEt_3_]^+^ may require no preliminary redox step.

On the other hand, the unconstrained DFT optimization of the Au(II)–Au(II) bimetallic system **H** also led to a substantial improvement of the coordination pattern, yielding an RMSD of only 0.23 Å (Table 2).

Hence, our DFT calculations indicate that Au(I)–Au(I) and Au(II)–Au(II) configurations are equally representative of the X-ray structure of NDM-1 treated with auranofin. The possible relationships between these two configurations have to be ascertained; however, we tentatively propose that Au(I)–Au(I) may be initially formed and that two-electron oxidation of the bimetallic scaffold may eventually lead to the Au(II)–Au(II) configuration. In this view, the Zn/Au exchange process, at least in its initial steps, is regarded as a non-redox process in which the supposed active species derived from auranofin, i.e., the [AuPEt_3_]^+^ cation, may react with the active site of NDM-1 via only ligand substitutions. Such a mechanistic hypothesis is the focus of ongoing studies.

The distribution of electron densities of the atoms forming the first coordination sphere of the Au–Au scaffold in the active site of NDM-1—systems **B**, **D**–**F**, and **H**—has been studied by means of Mulliken and NBO analyses (Table 3 and Appendix A). The high correlation of the two analyses’ data allowed us to better limit the discussion to only the NBO results (Table 3). In the case of system **B**, a triplet with a charge of -3, we found one unpaired electron at the center Au2, whereas the other one was detected half on the Au2-bound sulphur atom and half on the Au1 center (Table 3). On the other hand, both systems **D** and **E,** triplet and quintet, respectively, locate most of the unpaired electrons on the metal center Au1 (Table 3). In all these systems **B**, **D,** and **E**, the S and hydride O atoms present significant spin densities; the atoms are envisioned to be easily exposed to the attack by radical species produced in the biological milieu. Hence, the spin density analyses suggest that the Au(0)–Au(0) systems **B**, **D**, and **E** should be rather poorly representative of the bimetallic scaffold Au-bound NDM-1. In the neutral doublet system **F,** the unpaired electron was detected mostly (0.82) at the center Au1 and two coordinated nitrogen atoms, whereas a residual (0.18) spin density was found on the bridging hydroxyl (Table 3). In the triplet system **H**, the Au1 center and bridging hydroxyl localize approximately one unpaired electron, with almost the same distribution found in the system **F**, whereas the other unpaired electron was detected on the Au2 center, mostly on the metal and S atoms (Table 3). These calculations showed that systems **F** and **H** host the higher extent of spin density within the bimetallic scaffold, mostly on metal centers and the sulphur atom, with only a residual amount on the bulk-exposed hydroxyl ligands. We repute that these data evidence the higher redox stability of the **F** and **H** systems, being less suitable to or more protected from the attack of radical species, and suggest that in principle, Au(I)–Au(I) and Au(II)–Au(II) are the most representative configurations of the Au-bound NDM-1 enzyme.

To better corroborate our conclusions and analyze the effect of the Zn/Au exchange on the solvent exposure of the NDM-1 active site, we calculated the per residue solvent accessible surface (SAS) in the case of Au- or Zn-bound NDM-1 (Table 4). It was concluded that the metal substitution induces only an overall slight decrease of the SAS of the catalytic site of NDM-1, slightly more pronounced in chain B than in chain A, −2.3% and −8.3%, respectively. Interestingly, an appreciable increase of the SAS of the hydroxyl ligand was instead detected in the structure of chain A compared to the Zn-bound NDM-1. These data, together with the appreciable spin density on the hydroxyl oxygen atom detected on systems **D** and **E**, corroborate the lower stability of these configurations compared to the Au(I)–Au(I) and Au(II)–Au(II) configurations modelled by systems **C** and **H**. In the frame of the presented computational data, we hypothesized that the experimentally detected chains A and B of the Au-bound NDM-1 may be in fact chemically related; the hydroxyl-bridged chain A structure is more likely Au(II)–Au(II), a triplet, as represented by system **H**, and is yielded by the oxidation of the chain B-like structure in which the bimetallic scaffold is presumably Au(I)–Au(I), a singlet, as represented by system **C**.

Another aspect that we attempted to address in this study was providing an explanation to the tetrahedral coordination of the gold metal centers. The conclusions about the higher consistency of the Au(I)–Au(I) and Au(II)–Au(II) configurations, which typically disclose coordination numbers < 4, reinforce the templating role of the protein environment in determining the coordination geometries retrieved in the chain A and B structures of gold-bound NMD-1. The positional constraint exerted by the NMD-1 backbone on the gold-bound residues probably shapes the coordination geometry and stabilizes specific configurations of the bimetallic scaffold, i.e., singlet Au(I)–Au(I) or triplet Au(II)–Au(II).

The Zn/Au exchange yielded by the reaction of auranofin with the NDM-1 enzyme can be envisioned as a process eventuating in the metal-by-metal replacement with no significant structural aberration of the enzyme core.

## 4. Summary

Density functional theory calculations were carried out to investigate the unusual tetrahedral coordination assumed by the gold ions in the bimetallic core of the gold-bound NDM-1 enzyme, which is produced upon the treatment of the tiled zinc-dependent beta-lactamase with the metallodrug auranofin. By testing several charge and multiplicity schemes in combination with constraining the positions of the coordinating residues on/off, we showed that Au(I)–Au(I) and Au(II)–Au(II) moieties mostly resemble the experimental X-ray structure of the bimetallic scaffold of gold-bound NDM-1.

The most plausible scenario for the auranofin-based Zn/Au exchange in NDM-1 would see the early formation of the Au(I)–Au(I) system, followed by an oxidation step affording the Au(II)–Au(II) species, which disclosed the highest resemblance to the X-ray structure. The backbone fold of the NDM-1 enzyme exerts a template effect on the coordination geometry of the bimetal Au–Au scaffold by favoring the tetrahedral coordination and probably also influences the higher stability of the triplet Au(II)–Au(II) configuration with respect to other assets.

Moreover, thanks to the computational data presented here, the theoretical investigation of the mechanism of the Zn/Au exchange yielded by the reaction of the [AuPEt_3_]^+^ cation with NDM-1 is currently ongoing.

## Figures and Tables

**Figure 1 pharmaceutics-15-00985-f001:**
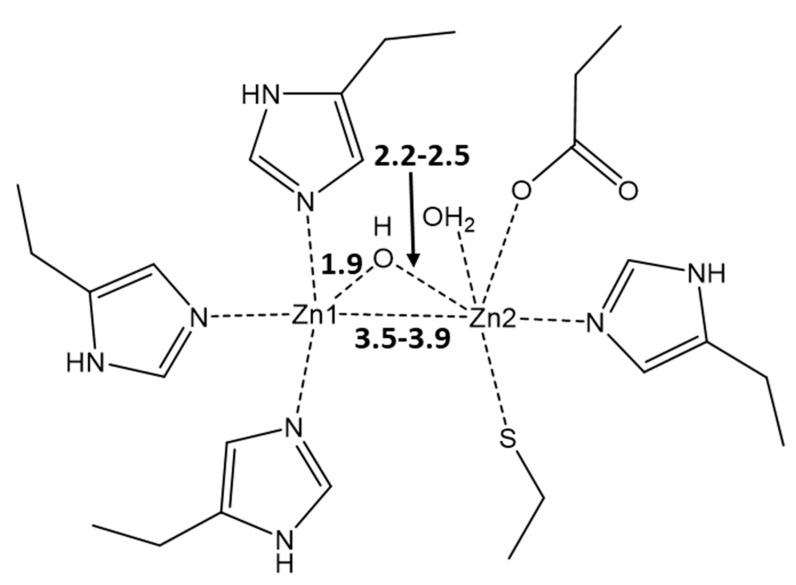
Typical active-site metal coordination geometry for B1 MBLs. The metal coordination and Zn–Zn distance are indicated with dashed lines. The typical distances between Zn1, Zn2, and OH are denoted in Å. The PDB code for the represented structure is 5zgz [12].

**Figure 2 pharmaceutics-15-00985-f002:**
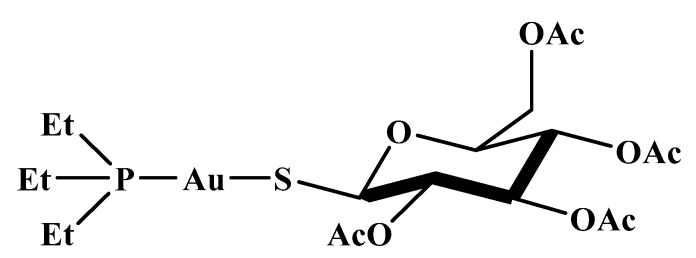
Auranofin’s structure.

**Figure 3 pharmaceutics-15-00985-f003:**
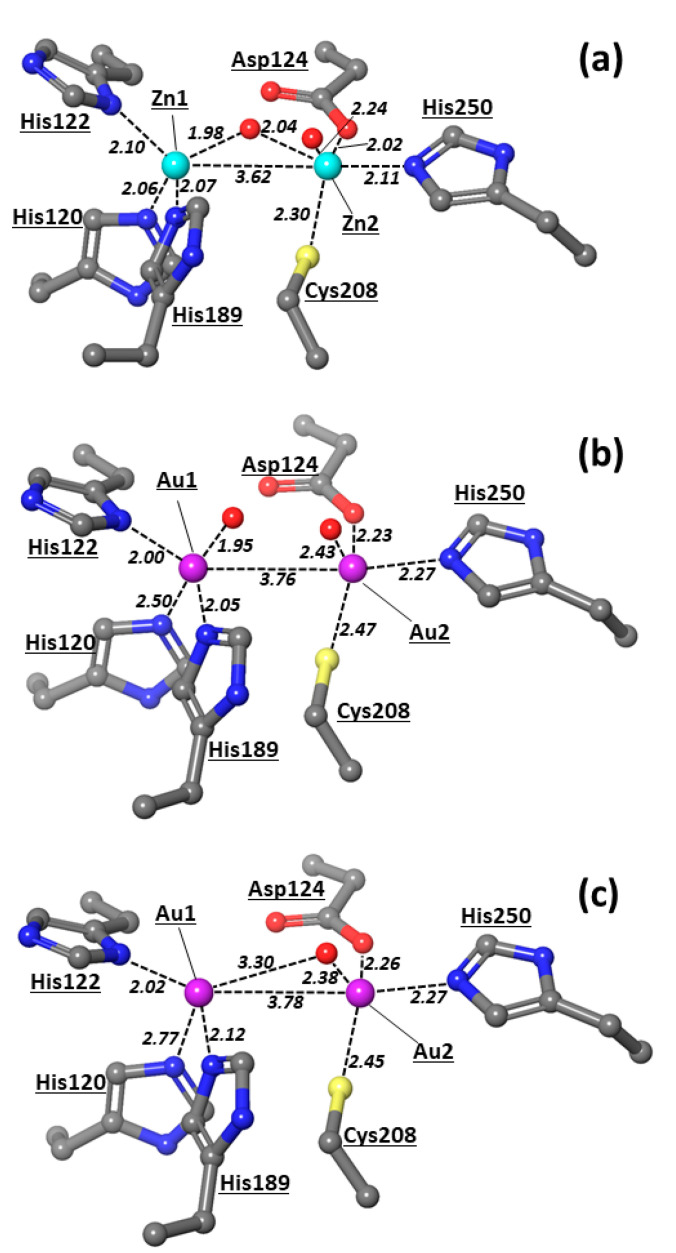
X-ray structures of the active site of NDM-1 with (**a**) Zn, (**b**,**c**) Au atoms. X-ray structures with Au2 coordinated to a water molecule ((**b**), chain A at PDB 6lhe) and without a water molecule in the first coordination sphere ((**c**), chain B at PDB 6lhe). Zn1/Au1 is always displayed at the left, and Zn2/Au2 is displayed at the right. The numeration of protein residues is taken from corresponding PDBs. All the distances are in Å and typed in italic font. The PDB codes for the represented structures are 5zgz [12] and 6lhe [15]. Color scheme employed in this figure and in other figures below: Zn (cyan), Au (plum), S (yellow), O (red), N (blue), C (grey), H (white).

**Figure 4 pharmaceutics-15-00985-f004:**
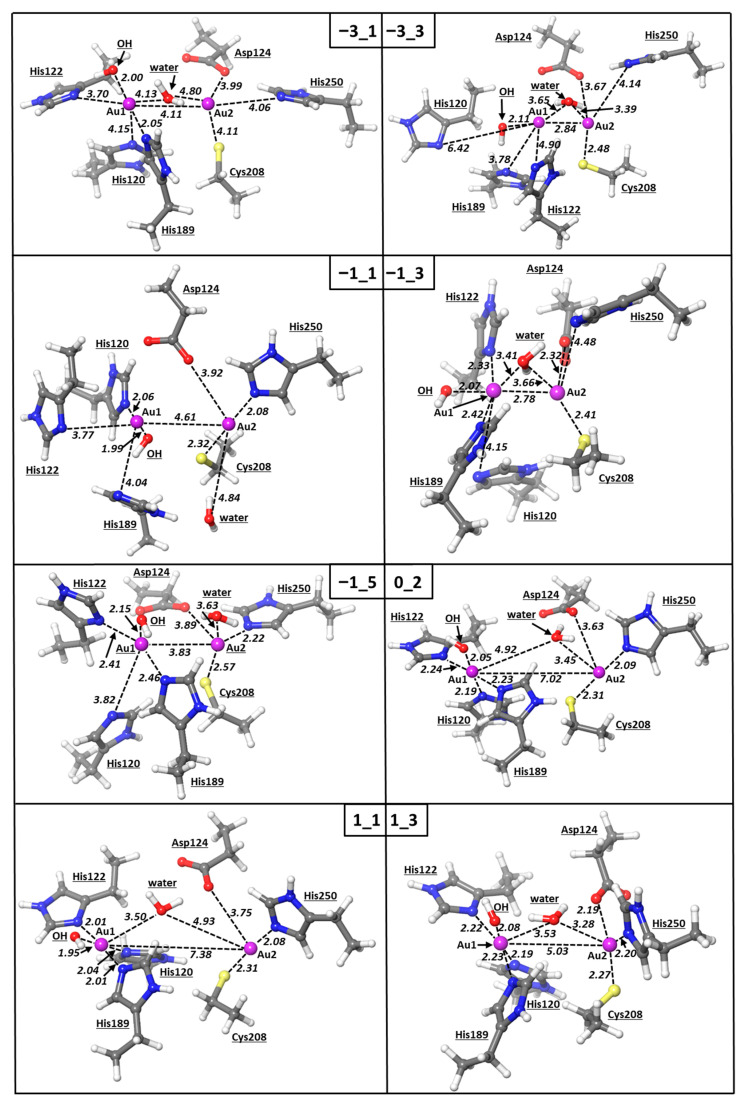
Optimized structures of the active site of NDM-1 with Au atoms. The nomenclature of the models is consistent with Table 1 (vide infra). The modelled NDM-1 residues are also reported. All the distances are in Å.

**Table 1 pharmaceutics-15-00985-t001:** The investigated bimetallic scaffolds at different charge–multiplicity combinations, named from **A** to **H**, with the corresponding electronic configurations and oxidation states.

Name of the Model	A	B	C	D	E	F	G	H
**Charge_Multiplicity**	-3_1	-3_3	-1_1	-1_3	-1_5	0_2	1_1	1_3
**Au-Au configuration**	s^2^d^10^–d^10^	s^1^d^10^–s^1^d^10^	d^10^–d^10^	s^1^d^9^–d^10^	s^1^d^9^–s^1^d^9^	d^10^–d^9^ s^1^d^10^–d^8^	d^10^–d^8^	d^9^–d^9^
**Au-Au oxidation state**	Au(0)-Au(0)	Au(0)-Au(0)	Au(I)-Au(I)	Au(I)-Au(I)	Au(I)-Au(I)	Au(I)-Au(II) Au(0)-Au(III)	Au(I)-Au(III)	Au(II)-Au(II)

**Table 2 pharmaceutics-15-00985-t002:** Comparison of the metal–metal and metal–ligand distances between the optimized structures of the **A**–**H** models (columns 2–8) and experimental data (last column). Last column demonstrates the RMSD values for computed distances with respect to the experimental data. The data computed for models **C**–**E** and **H**, disclosing the minimum RMSD with respect to the experimental data, are reported in the format “restrained optimization data/unrestrained optimization data”.

Distance		A	B	C	D	E	F	G	H	Exp *
**Au1**	Au2	4.11	2.84	4.61/3.55	2.78/2.62	3.83/5.96	7.02	7.38	5.03/4.30	3.76
**Au1**	N(His122)	3.70	4.90	3.77/3.54	2.33/2.39	2.41/2.45	2.24	2.01	2.22/2.29	2.00
	N(His189)	2.05	3.78	4.04/4.49	2.42/2.35	2.46/2.41	2.23	2.01	2.23/2.26	2.05
	N(His120)	4.15	6.42	2.06/2.04	4.15/3.96	3.82/3.70	2.19	2.04	2.19/2.16	2.50
	O(OH)	2.00	2.11	1.99/2.03	2.07/2.10	2.15/2.11	2.05	1.95	2.08/2.06	1.95
	O(H_2_O)	4.13	3.65	6.03/3.47	3.41/4.58	3.80/7.04	4.92	3.5	3.53/4.41	3.57
**Au2**	S(Cys208)	4.11	2.48	2.32/2.31	2.41/2.52	2.57/2.57	2.31	2.31	2.27/2.55	2.47
	N(His250)	4.06	4.14	2.08/2.08	5.25/2.31	2.22/2.19	2.09	2.08	2.20/2.06	2.27
	O(Asp124)	3.99	3.67	3.92/3.66	2.32/3.61	3.89/3.55	3.63	3.75	2.19/2.09	2.23
	O(OH)	5.20	4.93	4.64/3.73	4.84/4.72	4.74/7.58	7.93	8.83	5.45/4.34	3.60
	O(H2O)	5.20	3.39	4.84/3.70	3.66/3.38	3.63/3.35	3.45	4.93	3.28/3.35	2.43
**RMSD**		1.51	1.80	**1.41/0.95**	**1.42/0.89**	**0.85/1.84**	1.77	2.12	**0.74/0.48**	

* Retrieved from the X-ray structure with Au2 coordinated to a water molecule (PDB 6lhe, chain A) [15].

**Table 3 pharmaceutics-15-00985-t003:** The NBO atomic spin densities on each gold metal center with the respective coordinating protein atoms and on the hydroxyl or water oxygen are reported for the systems with multiplicities >1, i.e., **B**, **D**–**F**, and **H**. All values are in a.u.

Atom	B	D	E	F	H
Au1	0.41	0.86	1.32	0.57	0.60
N(His122)	0.00	0.14	0.14	0.12	0.13
N(His189)	0.00	0.13	0.15	0.12	0.12
N(His120)	0.00	0.00	0.00	0.06	0.08
Au2	0.96	0.32	1.09	0.00	0.44
S(Cys208)	0.35	0.16	0.67	0.00	0.41
N(His250)	0.01	0.00	0.19	0.00	0.15
O(Asp124)	0.01	0.05	0.01	0.00	0.06
O(OH)	0.20	0.31	0.38	0.18	0.17
O(water)	0.03	0.03	0.02	0.00	0.00

**Table 4 pharmaceutics-15-00985-t004:** Per residue solvent accessible surface (SAS) of the portion within 4.0 Å around the bimetallic system of the gold-bound and zinc-bound NDM-1 protein extracted from the pdb entries 6lhe and 5zgz, respectively. The M1/M2 residues correspond to either Au1/Au2 or Zn1/Zn2 metal centers; the OH and H_2_O residues correspond to the hydroxyl ion and water molecule coordinated to the bimetallic system. The total SAS and per atom SAS, i.e., total SAS divided by the number of non-hydrogen atoms, of the analyzed NDM-1 portions are reported in the last two rows. All values are in Å.

Residue	Gold-Bound		Zinc-Bound
	Chain A	Chain B	
His120	172.0	157.0	169.1
His122	140.8	136.9	162.3
Asp124	149.2	134.2	141.7
His189	169.7	155.0	173.5
Cys208	127.5	131.9	117.7
His250	225.0	214.4	243.5
M1	0.0	0.0	0.0
M2	0.0	0.0	0.0
OH	10.0	-	0.0
H_2_O	20.1	13.4	20.1
total SAS	1004.3	942.8	1027.9
per atom SAS	17.6	16.3	18.4

## Data Availability

Not applicable.

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
