# Peer review of "Auranofin Targeting the NDM-1 Beta-Lactamase: Computational Insights into the Electronic Configuration and Quasi-Tetrahedral Coordination of Gold Ions"

_pharmaceutics, 2023, doi:10.3390/pharmaceutics15030985_

Round 1

Reviewer 1 Report

The manuscript “Auranofin targeting the NDM-1 beta-lactamase: Computational insights into the electronic configuration and quasi-tetrahedral coordination of gold ions” present data concerning the computational modulation of gold interaction with active sites of NDM-1 beta-lactamase, aspect important in reversing the resistance acquired by pathogenic microorganisms containing such enzymes. My overall comment is that the data present interest considering the developing resistance to b- lactams antibiotics, most frequent used for infection treatment. Hence, the study start from the known binuclear Zn(II) active site of NDM-1 beta-lactamase and discusses the modifications associated with metal ions substitution with gold from auranofin. As result, this study is important in order to find modalities to bypass the resistance and improve the treatment of microbial infections.

I therefore recommend minor revision having in view the following aspects:

-        The hydroxide must be replaced by hydroxyl in whole paper. Hydroxide is an inorganic compounds having OH- in structure while this group (including coordinated one) is called hydroxyl or hydroxy group.

-        The tetragonal must be replaced by tetrahedral. The tetragonal comes from a octahedral distorted stereochemistry with coordination number 6 and not 4 such as for a tetrahedral stereochemistry.

-        The oxidation state for metal ions must be provided in parenthesis (i.e. Zn(II) instead of ZnII).

-        The oxidation state must be provided in all places were appear the symbol of a metal (i.e. Au(I) instead of Au) and when appear without metal must be provided with sign plus (i.e. +1 instead of I).

-        The expression “the chelation of zinc by complexes such as aspergillomarasmine A” must be modified as “the chelation of zinc by ligands such as aspergillomarasmine A” (page 3).

Author Response

Reviewer 1

… As result, this study is important in order to find modalities to bypass the resistance and improve the treatment of microbial infections.

I therefore recommend minor revision having in view the following aspects:

-        The hydroxide must be replaced by hydroxyl in whole paper. Hydroxide is an inorganic compounds having OH- in structure while this group (including coordinated one) is called hydroxyl or hydroxy group.

>> We thank Reviewer 1 very much for their recommendations. “Hydroxide” is substituted by “hydroxyl” in the manuscript.

-        The tetragonal must be replaced by tetrahedral. The tetragonal comes from a octahedral distorted stereochemistry with coordination number 6 and not 4 such as for a tetrahedral stereochemistry.

>> Done.

-        The oxidation state for metal ions must be provided in parenthesis (i.e. Zn(II) instead of ZnII).

>> Done.

-        The oxidation state must be provided in all places were appear the symbol of a metal (i.e. Au(I) instead of Au) and when appear without metal must be provided with sign plus (i.e. +1 instead of I).

>> Done in all cases but in occurrences like “Zn/Au exchange” and when the atom symbol is used to label an NDM-1 metal center, e.g. Zn1, Zn2, Au1, Au2.

-        The expression “the chelation of zinc by complexes such as aspergillomarasmine A” must be modified as “the chelation of zinc by ligands such as aspergillomarasmine A” (page 3).

>> Done.

Reviewer 2 Report

The manuscript "Auranofin targeting the NDM-1 beta-lactamase:
Computational insights into the electronic configuration and quasi-tetrahedral coordination of gold ions" presents a DFT study related to the Au-Zn metathesis in NDM-1 beta-lactamase enzyme. The main contribution of the authors in this paper is only the elucidation of geometrical parameters of the Au-based core protein fragment, thus confirming the main results of experimental data reported in Ref. 26.

So that, I consider that the novelty of this work is not so great, the main motivation starting from the cited paper in Ref. 26.

Other observations:

-The introduction is too long, the second paragraph should be removed, also the Fig 1.

-In Fig. 2 the carboxylate coordination at Zn2 ion must be corrected, the source of the metallo-enzyme core from PDB library must be added in legend.

-Since the Fig. 3 shows the PDB crystal structures of the NDM-1 core, so these are already confirmed, why authors choose to study these structures? Some exchange mechanisms are already published and these paper are also cited.

-In Results and discussion Section there are some discussion regarding the coordination environment of Au resulting from Au-Zn metathesis with references to metallo-core described in Fig. 3, already existent in PDB. The optimized structures also are in good concordance with the X-Ray structure of  Au-NDM-1 protein.

Based on these remarks my recommendation is Major revision.

Author Response

Reviewer 2

So that, I consider that the novelty of this work is not so great, the main motivation starting from the cited paper in Ref. 26.

>> We thank Reviewer 2 for their valuable comments which helped greatly to improve the manuscript. We believe that this work will be interesting to the readership of Pharmaceutics due to several reasons. Primarily, the mechanistic description of the Zn/Au substitution occurring in the active site of NDM-1, which is at the core of bacterial machinery responsible for degrading beta-lactam antibiotics, is novel and extremely interesting. Indeed, by a chemical point of view, the ligand exchange steps necessary to afford the final substitution of Zn(II) with Au(I) in the NDM-1 core is rather intriguing. Furthermore, this work sheds light onto the modus operandi of auranofin, one of the most prominent metallodrugs, the modern usage of which spans cancers, parasitic, bacterial, and neurodegenerative infections. Finally, we investigated the exotic quasi-tetrahedral coordination of Au centers that emerges from the combination of the oxidation states of gold ions with the shape of the bimetallic core of NDM-1. The above mentioned structural aspects may provide for a basis of data allowing the newly design of Au(I)-based antibacterial complexes.

Other observations:

-The introduction is too long, the second paragraph should be removed, also the Fig 1.

>> We thank this reviewer for this suggestion that allowed us to better structure the Introduction. Indeed, we removed the second paragraph as suggested, and moved below the first paragraph, right after the paragraph ending with “… directed at the β-lactam breaking.” (p.2) With this slight change, the more general “Bacterial resistance” concept is introduced before the more specific “Auranofin drug”.   

-In Fig. 2 the carboxylate coordination at Zn2 ion must be corrected, the source of the metallo-enzyme core from PDB library must be added in legend.

>> In former Figure 2 (Figure 1 in the current version), Zn2 ion is coordinated only to one oxygen of carboxylate, it is corroborated by the Zn-O distances of 2.02 and 3.36 Å. That is why the coordination only to one oxygen of carboxylate is shown. The PDB code was added to the legend.

-Since the Fig. 3 shows the PDB crystal structures of the NDM-1 core, so these are already confirmed, why authors choose to study these structures? Some exchange mechanisms are already published and these paper are also cited.

>> To the best of our knowledge, the Zn/Au exchange mechanism occurring in the NDM-1 bimetallic core has been never studied computationally at an atomistic detail. Indeed, the reference [Sun, H.; Zhang, Q.; Wang, R.; Wang, H.; Wong, Y.T.; Wang, M.; Hao, Q.; Yan, A.; Kao, R.Y.; Ho, P.L.; Li, H. Resensitizing carbapenem-and colistin-resistant bacteria to antibiotics using auranofin. Nat. Comm. 2020, 11(1), 1-3. https://doi.org/10.1038/s41467-020-18939-y] cites the article in which it was found experimentally that the Zn/Au exchange takes place and the resulting NDM-1 core with two Au ions was recorded via XRD, yet this experimental study has not disentangled the substitution mechanism itself.

 -In Results and discussion Section there are some discussion regarding the coordination environment of Au resulting from Au-Zn metathesis with references to metallo-core described in Fig. 3, already existent in PDB. The optimized structures also are in good concordance with the X-Ray structure of  Au-NDM-1 protein.

>> Indeed, the optimization was done with the density functional ωB97X which is known to yield geometrical structures and electronic energies with high accuracy.

Reviewer 3 Report

In the submitted manuscript the Authors have used DFT calculations to model the few systems differing in the total charge and multiplicity. The modelled systems were the “cage” systems presenting the interactions between histidines, Au atoms and water molecule. The amount of new information resulting from this study is therefore limited. Besides, there are other issues that should be solved, they’re presented below.

The abstract is presenting too much introduction-type information. It should be more focused on this current study and the obtained results.

Continuous line numbering would really facilitate the review process.

“Enterobacteriaceae, the presence of which in human bloodstream leads to the death in almost half of cases”-there is no verb in this sentence.

Are there any more cases of tetrahedral Au? Some other examples should be mentioned in the introduction. Are those cases limited to bimetallic systems?

Introduction is significantly too long, especially the part describing the MBLs. Especially, since some of those information are repeated in the Results section.

There are some benchmark works showing that the TPSS provides more accurate results on the gold compounds than wB97X.

My major problem in this study is that the Authors limit the whole active site to just a few residues, neglecting the role of the rest of protein. In such cases the ONIOM approach is usually applied. Why the Authors have not decided to do the ONIOM calculations?

Whole Page 6 is actually the description of the already recorded crystal structure.

Do all of the models (A-H) have the same initial (non-optimized) geometries with the only differences being charge and multiplicity? If yes, it should be clearly stated.

 Table 2, a similar table showing the differences between the calculated and experimental values should be created.

Author Response

Reviewer 3

The amount of new information resulting from this study is therefore limited.

>> We appreciate the comments of this reviewer. We believe that the Pharmaceutics readership could be extremely interested in the computational outcomes that provide for atomistic introspections of metal-based chemical systems, especially in the bioinorganic field, and may pave new addresses of investigations. The presented energy data draw a preliminary insight of the Zn/Au exchange thermodynamics that will be better addressed in a future paper (in preparation).

The abstract is presenting too much introduction-type information. It should be more focused on this current study and the obtained results.

>> Following this recommendation, the abstract was rewritten.

Continuous line numbering would really facilitate the review process.

>> We have added it.

“Enterobacteriaceae, the presence of which in human bloodstream leads to the death in almost half of cases”-there is no verb in this sentence.

>> Corrected (p. 1).

Are there any more cases of tetrahedral Au? Some other examples should be mentioned in the introduction. Are those cases limited to bimetallic systems?

>> The following text was added to the introduction (pp. 3-4), right after the sentence ending with “…geometries of gold ions are rather unusual.”, as suggested by this reviewer.

Although the preferred geometry of gold(I) complexes is linear with two-coordination at the gold center, instances of tetrahedral coordination have been reported [20]. Solution NMR studies have demonstrated that addition of excess phosphine to [AuL2]+ (L = phosphine) can result into the formation of the [AuL3]+ and [AuL4]+ species, but unexpectedly few of these complexes have been structurally characterized [21]. Furthermore, the majority of the four-coordinate species that have been identified comprise either bidentate phosphines [22] or thiolate ligands [23].

Introduction is significantly too long, especially the part describing the MBLs. Especially, since some of those information are repeated in the Results section.

>> We deleted text from the paragraph of introduction starting with “MBLs are zinc metalloenzymes, which feature one or two zinc ions” (pp. 1-2). We deleted:

One metal center, hereafter called Zn1, is coordinated to three histidine residues, while the other one, Zn2, is bound to Asp, Cys, His and one water molecule, whereas the bridging hydroxyl is stabilized by both metal centers (Figure 2). The crystal structures of NDM-1 disclose the usual geometry of the Zn1 and Zn2 metal centers as tetrahedral and trigonal bipyramidal, respectively, with the Zn-Zn distance varying in the range 3.5-3.9 Å [12, 18, 19]. The absence of acidic pKa [19, 20] and the short Zn1-O distance of 1.9 Å [19] allow to infer the hydroxyl nature of the bridging O at the X-ray structures. Moreover, it was shown that hydroxyl is bound to Zn2 much more weakly, as corroborated by the distance 2.2-2.5 Å between Zn2 and hydroxyl [12].

There are some benchmark works showing that the TPSS provides more accurate results on the gold compounds than wB97X.

>> The main reason for the employment of ωB97X is the perspective computational investigation of the mechanism of Zn/Au exchange that will be performed by using this functional. Indeed, we have often employed ωB97X to study several metal-based systems, and detected high accuracy in the estimate of the molecular geometry.

My major problem in this study is that the Authors limit the whole active site to just a few residues, neglecting the role of the rest of protein. In such cases the ONIOM approach is usually applied. Why the Authors have not decided to do the ONIOM calculations?

>> We thank this reviewer for her/his comment. In fact, as mentioned in the last paragraph of the summary, a further, mechanism-oriented investigation is going to be performed by us to address thermodynamics and kinetics of the Zn(II)/Au(I) exchange process. As correctly argued by this reviewer, the ONIOM approach would allow to include the electrostatic and steric effects exerted by the protein on the bimetallic core, and we are indeed planning to perform ONIOM calculations, presumably within the QM/MM scheme. On the other hand, as the reviewer knows, ONIOM calculations are particularly challenging, especially when dealing with the cutting of the system, and the low convergence of geometry optimization. Therefore, a more comprehensive inclusion of the steric effects would be rendered by sampling the phase space of the protein structure (both Zn-bound and Au-bound) via classical MD simulations. The presence of the whole NDM-1 structure in the model is essential when the ligand exchange events, occurring in the proposed mechanism of Zn(II)/Au(I) exchange, are focused. The computational study presented in this manuscript led to a first structural insight of the reactive system that might be assumed to be preliminary with respect to the forthcoming studies, including MD and QM/MM calculations. Indeed, we included all the residues within 3.5 Å distance from either of the two metal centers, i.e. all the coordinating residues, and kept the coordinate of C-alpha frozen. Such a constraint approximated the steric effect of the protein structure of Zn-bound and Au-bound NMD-1. We repute that such a set of constraints is adequate to simulate the anchoring of coordinative residues around the Au-Au bimetallic moiety imposed that the NDM-1 structure, and allow to assign the oxidative state of the two Au ions.

Whole Page 6 is actually the description of the already recorded crystal structure.

>> The description of the bimetallic moieties’ coordination retrieved from X-ray data is essential to start up the presentation of the computational work and data. On the other hand, we recognize that narrative can be improved to be slightly more concise. The modified narrative (pp. 5-6) is reported below:

The active site of NDM-1 is characterized by the presence of two zinc metal centers, Zn1 and Zn2, at a distance of 3.62 Å, and connected through a hydroxyl bridge (Figure 3a). The other ligands at Zn1 are three histidines 120, 122, and 189, that complete the tetrahedral coordination of this center. On the other hand, Zn2 reaches a trigonal bipyramidal geometry,through the coordination of Asp124, Cys208, His250, and a water molecule at the distances in the 2.2-2.5 Å range. Such a coordinative asset corroborates previous studies [9-11] that have indicated Zn1 as the metal center that affixes the OH in the correct position, whereas the labile water on Zn2 can be replaced by carboxylate substrates, for instance, the C3/C4 carboxylate of the β-lactam antibiotic [46].

The substitution of Zn(II) by Au ions [15], determines appreciable modifications of the NDM-1 active site (Figures 3b and 3c). Xray studies have led to two coordinative variants of the bimetallic gold moieties. In one case, two oxygens are bound to the Au1-Au2 scaffold; one oxygen is bound at a short distance (1.95 Å) with Au1, thus presumably being a hydroxyl group, whereas another oxygen is coordinated at Au2 at a longer distance (2.43 Å), thus more likely corresponding to a water ligand. The distance between the two Au centers is 3.76 Å (Figure 3b), which is longer than those detected in various crystallographic studies in the range 2.47-3.49 Å (Table S1) [47-54]. These data suggest that gold metal ions are not bound and held in place by the surrounding protein residues. Therefore, the formation of metallophilic Au(I)…Au(I) interactions – also denoted as closed-shell d10-d10 interactions – has been indicated to occur in the range 2.75-3.25 Å [55], so that the two gold centers observed in the NDM-1 enzyme are not directly bonded.

Beside the two oxygens bound at Au1 and Au2 metal centers, Au1 is also bonded to the three histidines 122, 189, and 120 at distances 2.00, 2.05, and 2.50 Å, respectively, while Au2 is bound to Asp124, Cys208, and His250 at distances of 2.23, 2.47, 2.27, respectively. Hence, both metal centers present a tetrahedral coordination, rather unusual for this transition metal species [56]. We can conclude that in the substitution of the Zn(II) ions with Au, Au1 resembles Zn1 in the coordination of the hydroxyl ligand, even though the latter group does not form a bridge as detected in the native NDM-1 structure. Therefore, the coordinative bond between Au1 and His120 resulted to be sensibly elongated compared to His122 and His189, thus distorting the tetrahedral-like coordination of Au1.

In the X-ray variant with just one oxygen (Figure 3c), the Au2-O distance of 2.38 Å is consistent with the coordination of a water molecule, while no hydroxyl group resulted coordinated to the bimetallic system. Such a reorganization is rather important because of the role played by the metal coordinated hydroxyl in the NDM-1-catalysis mechanism. On the other hand, we noticed that the coordination of Au1 shown in Figure 3c also features an increased elongation (by 0.27 Å) of the His120-Au1 bond compared to the system with the bound hydroxyl (Figure 3b).

Do all of the models (A-H) have the same initial (non-optimized) geometries with the only differences being charge and multiplicity? If yes, it should be clearly stated.

>> In fact, statements indicating that the same X-ray structure was used as input of all geometry optimization runs are already present in the manuscript:

p. 4: “We extracted a reduced system modelling the coordination of the gold bimetallic scaffold, incorporating two gold centers, their ligands, and the bridging hydroxyl, from the available X-ray structure [15], and assumed all possible combinations of charge and multiplicity of this system.”

p. 7: “Afterwards, we have conducted optimizations by starting from the same X-ray input geometry.” We added the word “same” now, to state it more clearly.

Table 2, a similar table showing the differences between the calculated and experimental values should be created.

>> In Table 2, the experimental data is reported in the last column, whereas the RMSD for the calculated distances with respect to experimental ones are reported in the last row of the table. The columns with the lowest RMSD values, i.e., the columns C, D, E, and H, are highlighted.

Round 2

Reviewer 2 Report

In the revised manuscript the authors added the required improvements in order to highlight the novelty and the significance of this subject in pharmaceutics. However, in Fig. 1 the coordination at Zn2 should be through C-O-Zn2, not C=O-Zn2, the carboxylate group being monodentate.

I recommend the publication of this paper after Minor revision.

Author Response

We thank Reviewer 2 for their accurate review. We have corrected the coordination of Zn2 in Figure 1.

Reviewer 3 Report

The corrected version is acceptable.

Author Response

We are very grateful to Reviewer 3 for their meticulous review.